# A prophage tail-like protein is deployed by *Burkholderia* bacteria to feed on fungi

Durga Madhab Swain [1], Sunil Kumar Yadav [1], Isha Tyagi [1], Rahul Kumar [1], Rajeev Kumar[1], Srayan Ghosh [1], Joyati Das[1] & Gopaljee Jha [1]

Some bacteria can feed on fungi, a phenomenon known as mycophagy. Here we show that a prophage tail-like protein (Bg_9562) is essential for mycophagy in *Burkholderia gladioli* strain NGJ1. The purified protein causes hyphal disintegration and inhibits growth of several fungal species. Disruption of the *Bg_9562* gene abolishes mycophagy. Bg_9562 is a potential effector secreted by a type III secretion system (T3SS) and is translocated into fungal mycelia during confrontation. Heterologous expression of Bg_9562 in another bacterial species, *Ralstonia solanacearum*, confers mycophagous ability in a T3SS-dependent manner. We propose that the ability to feed on fungi conferred by Bg_9562 may help the bacteria to survive in certain ecological niches. Furthermore, considering its broad-spectrum antifungal activity, the protein may be potentially useful in biotechnological applications to control fungal diseases.

[1] Plant Microbe Interactions Laboratory, National Institute of Plant Genome Research, Aruna Asaf Ali Marg, New Delhi 110067, India. Durga Madhab Swain, Sunil Kumar Yadav, Isha Tyagi and Rahul Kumar contributed equally to this work. Correspondence and requests for materials should be addressed to G.J. (email: jmsgopal@nipgr.ac.in)

**B**acteria co-evolve with other organisms and exhibit diverse interactions ranging from mutualism, commensalism, antagonism, and parasitism[1–7]. Due to shared environmental niche and nutrient constrain, inter-species competition occurs between co-habiting bacteria and fungi[5, 8]. Several bacteria demonstrate antifungal properties by producing antifungal metabolites, chitinolytic enzymes, siderophores, toxins, etc[9–12]. In addition, some of them can grow and multiply at the cost of living fungal biomass, utilizing fungi as source of nutrient and energy. This phenomenon is known as bacterial mycophagy[13–15]. For example, Collimonas bacteria can grow in autoclaved soil in the presence of actively growing fungal mycelia as sole carbon source[13, 16, 17].

Mycophagous bacteria can produce different types of antibiotics, toxins (tolaasin, lipase), cell wall degrading enzymes (chitinase, glucanase, and protease), and antifungal secondary metabolites, which are thought to damage fungal cells and release fungal metabolites to utilize them as nutrient source[14, 17]. Recently, through genomics and transcriptomics based approaches, efforts had been made to identify potential mechanisms involved in bacterial mycophagy[18, 19]. Furthermore, mycophagous bacteria could be the source of novel antifungal molecules, which are urgently required[20, 21].

To interact with other organisms, bacteria can use dedicated protein secretion systems to secrete proteinaceous effectors into the extracellular milieu or to deliver them into the cells of interacting organisms. The bacterial type III secretion system (T3SS) is one of the best characterized amongst them. It acts as a multiunit complex nanomachine to translocate effector proteins across the bacterial membrane to deliver them directly into eukaryotic host cells[22–26].

For example, the endofungal bacterium *Burkholderia rhizoxinica* utilizes its T3SS to establish a symbiosis with the fungus *Rhizopus microsporus*[27]. However, a role for T3SSs during mycophagous interactions has not been established yet.

We have recently isolated a bacterium with antifungal properties (*Burkholderia gladioli* strain NGJ1) from rice seedlings[28]. Here, we show that *B. gladioli* strain NGJ1 exhibits mycophagous activity on *Rhizoctonia solani* (causal agent of sheath blight disease in rice[29]) and other phytopathogenic fungi. Our results indicate that a prophage tail-like protein and a T3SS are required for this mycophagous ability.

## Results

**_B. gladioli_ strain NGJ1 demonstrates mycophagy.** The *B. gladioli* strain NGJ1 (here onwards referred as NGJ1) demonstrated antifungal activity on *Rhizoctonia solani*[28]. After a week of confrontation with *R. solani*, NGJ1 was found growing over the fungal biomass (Fig. 1a, Supplementary Fig. 1a). Under normal growth conditions, *R. solani* produces hundreds of viable secondary sclerotia on Potato Dextrose Agar (PDA) plates. However upon bacterial confrontation, only a few sclerotia were produced and they were unable to germinate when transferred onto either fresh PDA or acidified PDA plates (Fig. 1b, Supplementary Table 1). The bacteria were found growing surrounding the sclerotia on PDA plates, but no bacterial growth was observed on acidified PDA plates. Overall, this suggested that the sclerotia collected from confrontation plates have lost viability as the control sclerotia could germinate on both PDA as well as acidified PDA plates (Fig. 1b).

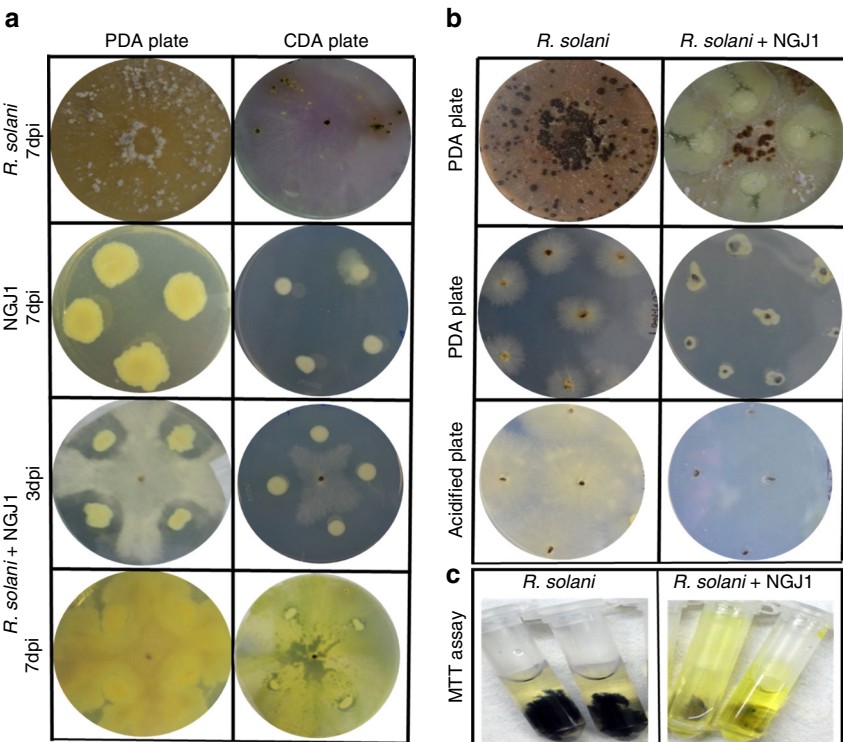

**Fig. 1** Mycophagous interaction of *B. gladioli* strain NGJ1 with *R. solani*. **a** Confrontation of NGJ1 with *R. solani* on PDA and CDA plates. Initially by 3 dpi, the bacteria showed antifungal activity while by 7 dpi, the bacteria were found growing over the entire fungal biomass. The growth patterns of *R. solani* and NGJ1 on both PDA and CDA plates are shown individually. **b** NGJ1 treatment inhibited the ability of *R. solani* to form secondary sclerotia. The sclerotia collected from NGJ1 confrontation plate (*R. solani* + NGJ1) failed to germinate on fresh PDA or acidified PDA plates while those collected from the untreated plates (*R. solani* alone) showed proper growth. **c** MTT staining of *R. solani* mycelia with and without NGJ1 treatment. The absence of color in bacterial-treated mycelia suggests cell death. Similar results were obtained in at least three independent experiments

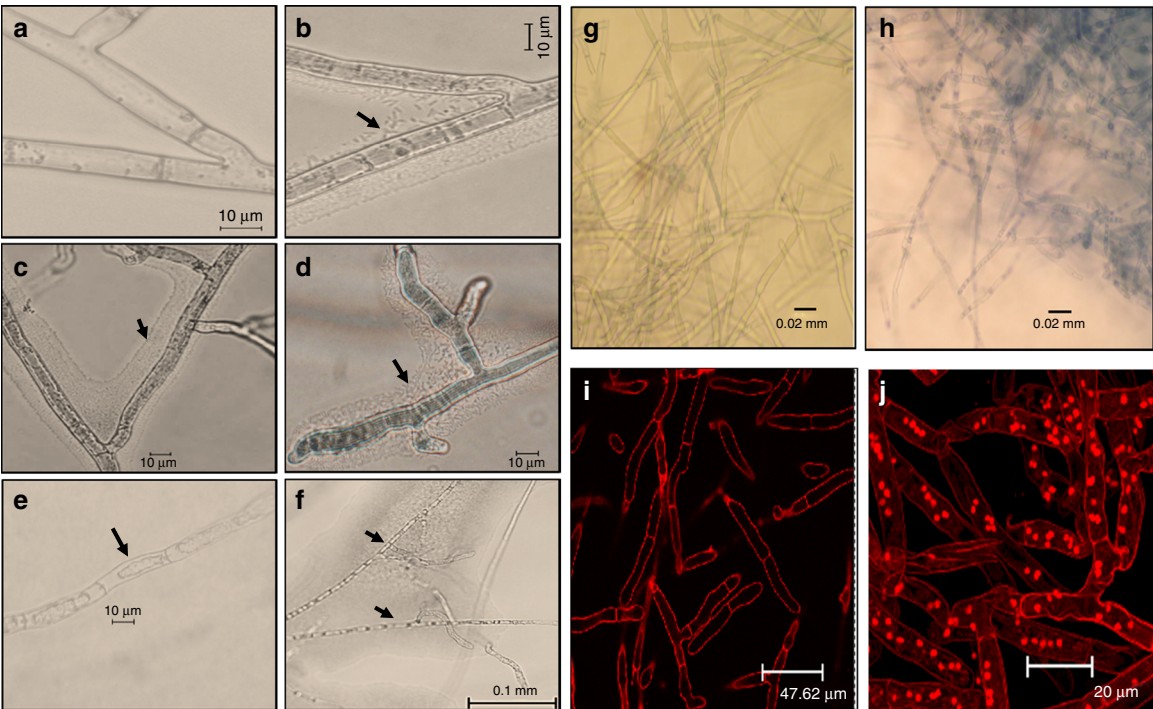

**Fig. 2** Treatment with *B. gladioli* strain NGJ1 induces cell death in *R. solani*. Microscopic view of **a** untreated and **b**–**f** NGJ1-treated mycelia. Majority of bacteria were found associated with fungal mycelia forming a thin film around the mycelial surface during 24 h of confrontation **b**, **c**. Alterations in mycelial integrity such as shrinkage of cytoplasm **e** and hyphae degeneration etc. were observed during 48–72 h of confrontation **d**, **f**. The untreated mycelia did not show any such alteration **a**. Microscopic images of trypan blue and propidium iodide (PI) stained NGJ1 untreated and treated *R. solani* mycelia. The uptake of trypan blue **h** and PI **j** stains in bacteria-treated mycelia suggested cell death while lack of trypan blue **g** and PI **i** staining in untreated mycelia indicates that they are alive. Similar results were obtained in at least three independent experiments

The inhibition of sclerotial germination was also observed when they were treated with NGJ1 culture ($10^9$ or $10^7$ cells/ml; Supplementary Fig. 2, Supplementary Table 2). However, when treated with diluted bacterial culture ($10^3$ or $10^5$ cells/ml), some of the sclerotia could initially germinate, but subsequently NGJ1 grew over them and prevented their further growth. Heat killed bacteria were unable to prevent sclerotial germination (Supplementary Fig. 2, Supplementary Table 2). We also tested the bacterial ability to prevent disease caused by *R. solani*. Notably, NGJ1-treated sclerotia failed to cause disease in rice as well as in tomato while the control sclerotia (without NGJ1 treatment) caused characteristic disease symptoms on both plants (Supplementary Fig. 3).

The interaction of NGJ1 and *R. solani* was also studied on Czapek Dox Agar (CDA), a semi-synthetic minimal media used for culturing fungi. In general, NGJ1 exhibited limited growth on CDA plates; however, upon contact with fungal mycelia its growth was enhanced (Fig. 1a). Similarly, drastic increase in bacterial growth was also observed when it was co-cultivated with *R. solani* mycelia in Czapek Dox Broth (CDB) media (Supplementary Fig. 1b).

Furthermore, we studied the NGJ1 and *R. solani* interactions through microscopic analysis. During 24 h of confrontation, large numbers of bacterial cells were found associated with fungal hyphae (Figs. 2b, c). Degenerated and highly septate hyphae exhibiting cytoplasmic shrinkage were observed in bacterial-treated mycelia during 48–72 h of confrontation (Figs. 2d–f). However, as shown in Fig. 2a, the control (untreated) mycelia were having intact hyphal integrity. The NGJ1 treatment was also found to induce cell death responses in pre-grown fungal mycelia, as revealed by trypan blue (Figs. 2g, h) and propidium iodide (PI) staining (Figs. 2i, j). The MTT

[3-(4,5-dimethylthiazol-2-yl)-2,5-diphenyltetrazolium bromide] assay further suggested that treated mycelia have lost viability, as they were defective in reducing MTT into a colored compound, i.e, formazon (Fig. 1c, Supplementary Fig. 4c). Moreover, it was observed that NGJ1 treatment not only suppressed growth of pre-grown mycelia but also caused reduction in fungal biomass (Supplementary Fig. 4a, b). These observations indicate that NGJ1 might be degrading fungal mycelia to feed on them.

In summary, our data demonstrate that *B. gladioli* strain NGJ1 exhibits bacterial mycophagy as it could grow and multiply at the cost of *R. solani* biomass. It is noteworthy that the mycophagous ability of NGJ1 is not restricted to a particular strain of *R. solani*. Mycophagy was also observed on different strains of *R. solani* and various other fungi including *Fusarium oxysporum*, *Magnaporthe oryzae*, *Venturia inaequalis*, *Ascochyta rabiei*, *Alternaria brassicae* as well as the oomycete *Phytophthora* sp., which has a non-chitinaceous cell wall (Supplementary Fig. 5).

**A T3SS is required for bacterial mycophagy.** We further explored the bacterial–fungal interaction to identify molecules involved in bacterial mycophagy. Generally the T3SS is known to play a crucial role during host-pathogen interactions[22], [23]. We observed that *B. gladioli* strain NGJ1 harbors a canonical injectosome subtype of T3SS (Supplementary Fig. 6). In order to test the involvement of T3SS in bacterial mycophagy, we disrupted the *hrcC* (WP_013689095.1) gene (the core component of T3SS apparatus[30]) through pGD4 plasmid integration and obtained two independent T3SS deficient mutants, namely NGJ12 and NGJ13 (Supplementary Table 4). Besides being defective in inducing hypersensitive response (HR) in *Nicotiana benthamiana* leaves, both the T3SS mutants were defective in

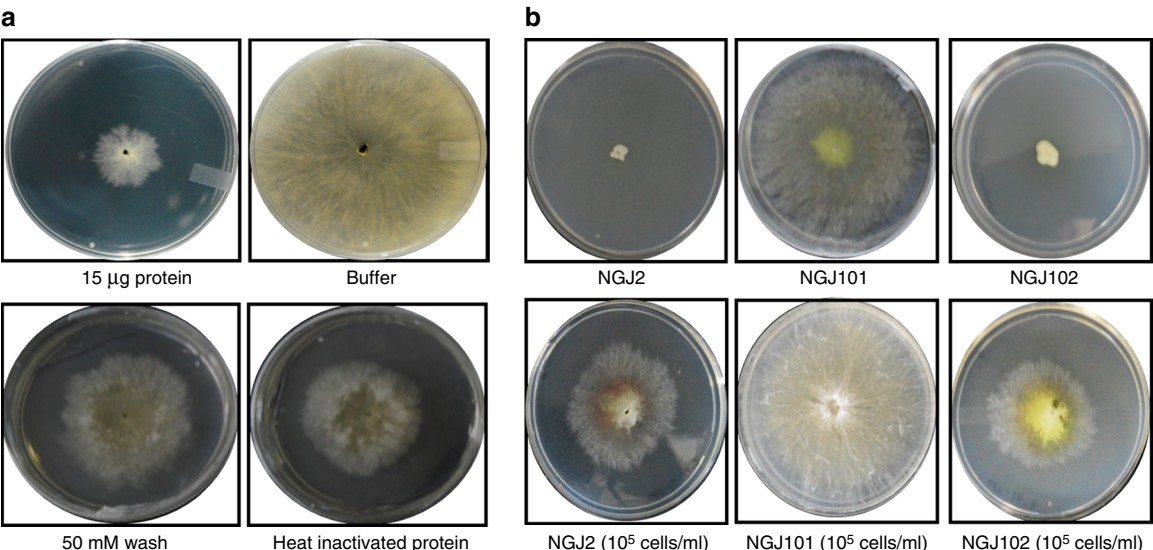

**Fig. 3** A prophage tail-like protein (Bg_9562) is required for bacterial mycophagy. **a** Effect of Bg_9562 protein (15 μg/ml) and various control treatments on sclerotial germination and growth of *R. solani* on fresh PDA plates at 48 h post treatment. The growth was found to be restricted upon protein treatment. **b** Effect of *Bg_9562* mutant and complement on sclerotial germination and growth. The mutant (NGJ101) was defective in preventing sclerotial germination while the wild type (rif[R] derivative of NGJ1; NGJ2) and complement (NGJ102) were proficient. Upon treatment with 10[5] cells/ml of bacterial cultures, the sclerotia could germinate initially but subsequently the NGJ2 as well as NGJ102 grew over fungal biomass while NGJ101 failed to do so. Similar results were obtained in at least three independent experiments

mycophagous ability as they were unable to forage over *R. solani* mycelia (Supplementary Fig. 7).

**Identification of a prophage protein as a potential T3SS effector.** Further using an online tool (www.effector.org), we predicted proteins encoded in the NGJ1 draft genome[28] which have potential T3SS secretion signal (effective T3 score 1)[31]. Out of 35 such proteins, the presence of one (Bg_9562) having homology to a phage tail protein was quite surprising. Phylogenetic and blastX analysis indicated that the Bg_9562 protein demonstrates significant sequence similarity with predicted phage tail proteins in different *Burkholderia* and *Paraburkholderia* species (Supplementary Fig. 8). The Bg_9562 as well as its various orthologs were found to harbor a conserved *phage_TAC_7* superfamily domain, generally associated with bacteriophage tail protein Gp41. Notably, no known toxic or lytic domain was detected in Bg_9562 and its orthologs. Interestingly the *Bg_9562* gene was found to be located in the middle of an apparent prophage (having sequence similarity with P2-like phages of *B. cepacia* complex[32]) gene cluster in different *B. gladioli* genomes, including NGJ1 (Supplementary Fig. 9). However, the cluster seems incomplete as genes encoding phage head assembly or capsid proteins are absent at this locus, suggesting that it cannot be induced to form an active phage. We also adopted various standard methodologies (phage plaque assay, phage particle isolation, scanning electron microscopic analysis, and induction of phage related genes through quantitative PCR) but could not detect phage induction.

**Bg_9562 is delivered into fungal mycelia in a T3SS-dependent manner.** We further endeavored to demonstrate the potential T3SS secreted nature of Bg_9562 protein. The Bg_9562 protein was overexpressed in *E. coli* using a pET28a bacterial overexpression system, purified, and confirmed through western blotting using anti-His-antibody (Supplementary Fig. 10a). The identity of purified protein was validated through nanoLC/MS-MS and Bg_9562 protein specific polyclonal antibodies were raised. Western blot analysis revealed that Bg_9562 protein is

produced during growth of NGJ1 in PDB broth. Upon confrontation, it gets translocated into *R. solani* mycelia as we could detect the protein in NGJ1-treated mycelia (Supplementary Fig. 11a). The T3SS mutants (NGJ12 and NGJ13) strains of NGJ1 were also able to synthesize Bg_9562 protein. But they were unable to deliver the protein into fungi, as the protein could not be detected in *R. solani* mycelia that were treated with T3SS mutants (Supplementary Fig. 11b). Overall, this suggests that Bg_9562 may be a T3SS effector and that during confrontation it gets translocated into *R. solani* mycelia in a functional T3SS-dependent manner.

**Bg_9562 demonstrates broad-spectrum antifungal activity.** We further tested the effect of purified Bg_9562 protein on fungal growth. Amongst various concentrations, the 15 μg/ml concentration of the protein was found to be efficient in suppressing sclerotial germination (Supplementary Fig. 10c). Restricted mycelial growth and reduced secondary sclerotia formation were observed when protein-treated sclerotia were grown on fresh PDA plates (Fig. 3a). However, the sclerotia treated with various controls (including PBS buffer, column wash, and heat inactivated protein) did not show any growth restrictions (Fig. 3a; Supplementary Fig. 10c). This indicates that treatment with the prophage tail-like protein (Bg_9562) inhibits *R. solani* growth. Moreover, treated mycelia failed to reduce MTT into formazon (colored compound) thus suggesting that the protein treatment had induced cell death (Supplementary Fig. 10b). Furthermore, restriction in *R. solani* growth was also observed when pre-grown mycelia on agar slides were exposed to the protein (Figs. 4a, b). In contrast, buffer-treated mycelia continued to grow and in-due course of time covered the entire agar slide (Figs. 4a, b). Microscopic analysis revealed deformation in protein-treated mycelia but not in control samples (Figs. 4c–f and Supplementary Fig. 12). The leakage of cytosolic components along with shrinkage of cell membrane and deflated hyphal structures were apparent in protein-treated mycelia (Figs. 4d–f and Supplementary Fig. 12g–i). Most importantly, the alteration in mycelial structure observed upon Bg_9562 protein treatment

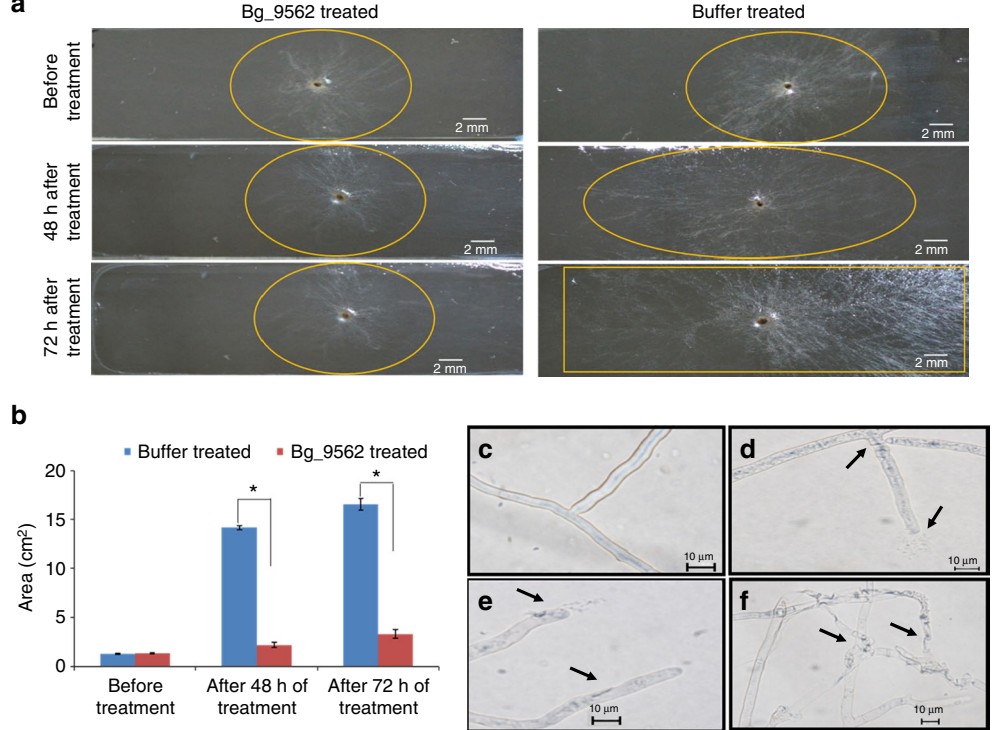

**Fig. 4** The Bg_9562 protein prevents mycelial growth and induces cell death responses in *R. solani*. **a** Effect of protein and buffer treatment on the growth of *R. solani* mycelia on agar slides. **b** Growth of buffer and 15 µg/ml Bg_9562 protein-treated *R. solani* mycelia on agar slides, measured in terms of observed growth area (cm$^2$). **c–f** Microscopic view of buffer **c** and 15 µg/ml protein **d–f** treated mycelia. Leakage of cytoplasmic components from hyphae **d**, **e** along with deflated mycelia **f** were some of the prominent effects of protein treatment. The arrows highlight mycelial damage and *asterisks* indicate statistical significant difference between indicated groups at $P < 0.001$ (estimated by one-way ANOVA). Graphs show mean values ± s.d. The experiments were repeated at least three times with minimum three technical replicates and similar results were obtained

was similar to that noticed following NGJ1 treatment (Supplementary Fig. 12). Moreover, it is worth-noting that purified protein had broad-spectrum antifungal activity. It inhibited the growth of *Saccharomyces cerevisiae*, *Candida albicans*, *Fusarium oxysporum*, *Ascochyta rabiei*, *Phytophthora* sp., *Alternaria brassiceae*, *Magnaporthe oryzae*, *Venturia inaequalis*, and other fungi (Supplementary Fig. 13). The quantitative effect of protein treatment on growth of various fungi is summarized in Supplementary Fig. 14. However, it is interesting to note that Bg_9562 protein demonstrates no adverse effect on bacterial growth (including *B. gladioli* strain NGJ1, *Pantoea ananatis*, and *E. coli*; Supplementary Fig. 15).

**Bg_9562 is required for mycophagy**. In order to probe the role of this protein in bacterial mycophagy, we disrupted the *Bg_9562* gene through integration of pGD2 plasmid (Supplementary Table 4). Two independently isolated mutants (NGJ100 and NGJ101) showed defects in bacterial mycophagy. Both of them failed to inhibit sclerotial germination and were incapable of foraging over fungal biomass during confrontation (Fig. 3b). The defect in mycophagous activity was complemented by expressing the full-length copy of *Bg_9562* gene through a broad host range plasmid (pHM1) in NGJ102 (Fig. 3b).

**Heterologous expression of Bg_9562 imparts bacterial mycophagy**. It is pertinent to note that type III secretion system (T3SS) of a bacterium can efficiently deliver the type III effectors of other bacteria. For example, *Xanthomonas campestris* pv. vesicatoria had been shown to deliver the effectors of other plant (*Ralstonia solanacearum*, *Pseudomonas syringae* pv. glycinea) and

mammalian (*Yersinea pseudotuberculosis*) pathogenic bacteria in a T3SS-dependent manner[33]. As *R. solanacearum* and *B. gladioli* are phylogenetically belonging to the same β-proteobacteria group, we tested whether the heterologous expression of Bg_9562 could impart mycophagous ability in *R. solanacearum* (a non mycophagous bacterium). For this we mobilized the pHM1 plasmid containing *Bg_9562* gene (pGD3) into a *R. solanacearum* strain F1C1[34]. The recombinant strain (F1C1N3) had gained mycophagous ability as it could forage over *R. solani* mycelia (Supplementary Fig. 16). Moreover, it was noteworthy that mobilization of pGD3 plasmid into a T3SS deficient strain (F1C1N2) failed to provide mycophagous ability in the recombinant *R. solanacearum* strain F1C1N4 (Supplementary Fig. 16). Furthermore, the growth rate of F1C1N3 was significantly enhanced in the presence of *R. solani* mycelia, while this effect was not found with either F1C1 or F1C1N4 (Supplementary Fig. 17). Together, these results confirm that heterologous expression of Bg_9562 imparts mycophagous ability to *R. solanacearum* and that a functional T3SS is required for the gain of such activity.

Overall, the data presented in this study revealed the role of *Bg_9562* gene encoding a prophage tail-like protein during bacterial mycophagy. It is known that in a lysogenic state, phages can confer advantageous traits to their hosts[35–39]. Lysogenic conversion had also been thought to be playing important role in the evolution and pathogenesis of *Burkholderia* species[40, 41]. It seems that the *B. gladioli* strain NGJ1 has evolved to occupy a specific ecological niche by utilizing a prophage tail-like protein as a potential T3SS effector to feed over fungi. By feeding on fungi, mycophagous bacteria might have a significant effect in shaping the diversity and dynamics of host associated

microbiomes. Additionally, by demonstrating the broad-spectrum antifungal activity of a *B. gladioli* prophage tail-like protein, our study opens up new possibilities wherein the protein can be explored for biotechnological applications in controlling fungal diseases of plants as well as animals.

## Methods

**Growth conditions**. The bacterium *Burkholderia gladioli* strain NGJ1 was grown on Potato Dextrose Agar (PDA; Himedia, India) plates at 28 °C. The *Pantoea ananatis* and *E. coli* (DH5α) were grown on PDA (at 28 °C) and LBA (Luria Bertani Agar; Himedia, India; at 37 °C) plates, respectively. *Ralstonia solanacearum* strains were cultured in BG media (1% Bacto-peptone, 0.1% Casamino Acids, 0.1% yeast extract and 0.5% glucose) and BG containing 1.6% agar was used for growth in solid media. The *Rhizoctonia solani* strain BRS1 was grown on PDA or CDA (Sucrose: 3%, NaNO$_3$: 0.2%, K$_2$PO$_4$: 0.1%, MgSO$_4$: 0.05%, KCl: 0.05%, FeSO$_4$: 0.001% and Agar: 1.5%; Himedia, India) plates at 28 °C. Various other fungi used in this study and their growth conditions are summarized in Supplementary Table 3. The fungal and bacterial strains used in this study are listed in Supplementary Tables 3 and 4, respectively.

**B. gladioli strain NGJ1-R. solani confrontation assays**. *R. solani* forms sclerotia, the spore-like resting structures. The sclerotia freshly collected from 2-week-old culture plate of *R. solani* AG1-IA strain BRS1 were placed at the center of PDA or CDA plates (using sterile toothpicks). Thereafter NGJ1 culture (20 µl of 10$^9$ cells/ml) was spotted at four corners of the plate, equidistant from the center. The plates were incubated at 28 °C and routinely monitored for confrontation. Once the bacteria established physical contact with fungal mycelia, bacterial abundance on mycelial lawn was estimated by serial dilution plating and colony counting at 4, 5, and 7 days post inoculation (dpi). Growth ability of the secondary sclerotia collected from control or bacterial confrontation plates were studied by placing them on fresh PDA as well as acidified PDA plates, using sterile toothpicks (Fig. 1b).

For confrontation in liquid media, fungal sclerotia were initially grown for 48 h in 10 ml of CDB (Himedia, India) media at 28 °C with constant shaking (200 rpm) to obtain mycelial mass. Thereafter, 1% of 10$^9$ cells/ml NGJ1 culture was inoculated into it and after different time intervals (0 h, 24 h, and 48 h) of co-cultivation, bacterial growth was estimated by dilution plating on PDA plates, followed by colony counting. As control, the bacterial growth in CDB media in absence of *R. solani* mycelia was similarly estimated at each time point.

Further to estimate the impact of bacterial treatment on fungal biomass, the *R. solani* mycelia were pre-grown in 10 ml PDB for 48 h. The pre-grown fungal mycelia were blotted on sterile filter disk to remove excess water and semi-dry weight (initial biomass) was measured aseptically. Thereafter the mycelia were inoculated with NGJ1 (1% inoculum of 10$^9$ cells/ml bacterial culture) or without NGJ1 (control; PDB) culture. After 48 h of subsequent growth at 28 °C, semi-dry weight (final biomass) of bacterial treated and untreated mycelia were again measured aseptically. The gain or loss in fungal biomass was estimated by subtracting initial biomass from the final biomass. The experiment was independently repeated at least three times with minimum three technical replicates.

**Sclerotial growth prevention assay**. The *R. solani* sclerotia ($n = 5$) were treated with 10 ml of different concentrations of NGJ1 culture (10$^9$, 10$^7$, 10$^5$, and 10$^3$ cells/ml). After 4 h of bacterial treatment at 28 °C, the sclerotia were taken out, washed thoroughly with sterile water, and placed on fresh PDA plates to grow at 28 °C. As control, the sclerotia were similarly treated with heat killed bacterial culture (by boiling 10$^9$ cells/ml NGJ1 culture for 1 h; at 100 °C) as well as PDB broth and analyzed for germination ability. Each experiment was repeated at least thrice with minimum 5 sclerotia per treatment being analyzed in each replicate.

**Agar slide confrontation assay and microscopic analysis**. Microscopic glass slides were washed with ethanol and autoclaved. They were overlaid with a thin layer of sterile 1% agar and these agar slides (without any added nutrients) were used for studying bacterial–fungal interactions. For this, the fungal sclerotia were placed at both corners of the slides while NGJ1 was patched at center. The slides were incubated at 28 °C under aseptic condition to monitor bacterial and fungal growth. After 24, 48, and 72 h of confrontation, slides were analyzed for hyphal morphology under Nikon 80i-epi-microscope. Only representative images have been presented. The experiment was repeated at least three times, each time minimum three technical replicates were analyzed.

**Cell death assays**. *Trypan blue staining*. The NGJ1 treated (1% inoculum of 10$^9$ cells/ml) and untreated pre-grown *R. solani* mycelia after 48 h of co-cultivation/growth in PDB media were stained with 1% trypan blue. Upon incubation for 10 min at room temperature, the mycelial mass was analyzed under Nikon 80i-epi-microscope to observe internalization of the stain.

*PI staining*. The NGJ1 treated (1% inoculum of 10$^9$ cells/ml) and untreated *R. solani* mycelia were stained with PI (2 mg/ml) and upon incubation in dark for

1 h, the stained mycelia were analyzed under ×63 objective of Leica TCS-SP2 Confocal Laser Scanning Microscope. A HeNe laser at 543 nm excitation and emission above 560 nm (LP) was used to detect PI internalization. Several images were taken at different thickness ranging from 15 to 20 µm and overlays were produced using the LAS AF Version: 2.6.0 build 7266 software.

*MTT assay*. NGJ1 (1% inoculum of 10$^9$ cells/ml) treated and untreated pre-grown *R. solani* mycelia after 48 h of co-cultivation/growth in PDB were harvested and washed thoroughly with PBS buffer (phosphate buffer saline; 10 mM, pH 7.4). 900 µl of PBS and 100 µl of MTT (3-(4,5-dimethylthiazol-2-yl)-2,5-diphenyltetrazolium bromide) solution (5 mg/ml suspended in PBS buffer) was added and incubated in dark for 90 minutes as per the protocol described[42]. The samples were washed with buffer to remove loosely attached MTT dyes and observed for appearance of dark-color, due to production of formazon. Further formazon was extracted with absolute ethanol by incubating overnight at room temperature and OD$_{570}$ was measured using spectrophotometer (Biorad Smart Spec-3000). The trypan blue, PI, and MTT assays were repeated with three biological and technical replicates.

**Bio-control assay**. The freshly grown *R. solani* sclerotia were treated with 5 ml (10$^9$ cells/ml) of NGJ1 culture for 4 h at 28 °C and inoculated into tillers of 45 days old rice cv. TP309 (*Oryza sativa* ssp. japonica; lab collection). *R. solani* sclerotia without NGJ1 treatment were also inoculated as control. The plants were incubated further in a PGV15 conviron plant growth chamber at 28 °C temperature, 80% relative humidity, and 16/8 h of day/night. The phenotypic observations were recorded at 3 and 5 days post pathogen inoculation (3 and 5 dpi). The relative vertical sheath colonization (RVSC) index on rice was calculated as described[43].

For infection of tomato plants, the NGJ1 treated and untreated *R. solani* sclerotia were placed over abaxial surface of detached leaves of tomato (*Solanum lycopersicum* cv. pusa ruby; lab collection) and incubated at 28 °C in petri-dish under moist condition. The leaves were analyzed for appearance of disease symptoms at 3 dpi. The experiment was repeated at least three times and each time minimum 15 rice tillers (3 pots; 5 tillers in each pot) as well as 15 tomato leaves were inoculated. Similar numbers of un-inoculated rice tillers or tomato leaves were used as negative control in each experiment.

**Confrontation of B. gladioli strain NGJ1 with other fungi**. The mycophagous ability of NGJ1 was tested on several other fungi (Supplementary Fig. 5). For fungi that grow at 28 °C, the confrontation assay was performed as described in case of *R. solani*. For other fungi (*Venturia inaequalis*, *Magnaporthe oryzae*, and *Ascochyta rabiei*), initially they were grown at 22 °C until sufficient mycelial lawn was formed on the plate. Thereafter the NGJ1 culture was spotted as described before. The plates were subsequently incubated at 28 °C and routinely observed for mycophagy.

**Confrontation of T3SS mutant B. gladioli strain with R. solani**. Partial fragment (300 bp) of *hrcC* gene (accession code WP_013689095.1) was PCR amplified from *B. gladioli* strain NGJ1 using hrcCF and hrcCR gene specific primers (Supplementary Table 4) and cloned into pK18 mob vector[44] to obtain pGD4 plasmid. The pGD4 plasmid was transformed into *E. coli* S17-1 and mobilized from S17-1 into NGJ2 (a spontaneous rif$^r$ derivative of *B. gladioli* strain NGJ1) through conjugation as per the method described earlier[45]. The insertion mutants were selected on kanamycin (50 µg/ml) and rifampicin (50 µg/ml) containing KBA (King's medium B Base; Himedia, India) plates. In this process, two independent Δ*hrcC* single recombinant insertion mutants namely NGJ12 and NGJ13 were raised. Both mutants were confirmed by PCR using gene specific flanking forward (hrcCf1) and vector specific reverse (M13 rev) primers. To further confirm, HR assay was performed using *Nicotiana benthamiana* (lab collections) leaves. For this, 24 h pre-grown inoculum (10$^7$ cells/ml) of mutant (NGJ12 and NGJ13) as well as wild type (NGJ2) bacterial strains were infiltrated into *N. benthamiana* leaves using a needle less syringe. The severity of HR was monitored and photographed at 48 h post infiltration. The confrontations of NGJ12, NGJ13, and NGJ2 with *R. solani* were performed as described above. The plates were incubated at 28 °C and photographed at different time intervals to visualize mycophagy. Further effect of lower concentration (10$^5$ cells/ml) of NGJ12, NGJ13, and NGJ2 bacterial strains on sclerotial growth prevention was also studied as per the methodology described above. Each experiment had been repeated at least thrice with minimum 5 sclerotia per treatment being analyzed, in each replicate.

**Computational analysis**. The presence of type III secretion system (T3SS) in different *B. gladioli* strains were analyzed by using *Burkholderia* genome database[46] (http://www.burkholderia.com). The predicted proteins of *B. gladioli* strain NGJ1[28] were mined using T3SS effector prediction database[31] (http://effectors.org). The proteins having size range 50–150 aa, with effective T3 score 1 and lacking canonical signal for sec dependent secretion were screened for further analysis. The blast annotation revealed one of them (Bg_9562) to have high sequence similarity with different phage tail proteins (Supplementary Fig. 8). The Bg_9562 and its orthologs in different *B. gladioli* strains were found located in a bacteriophage genomic cluster, when searched in *Burkholderia* genome database[46]. Phylogenetic analysis was conducted in MEGA6 tool[47] using neighbor-joining method[48]. To identify presence of conserved domain, the NCBI conserved domain database

search[49] was performed using amino acid sequence of the Bg_9562 as well as its various orthologs. The nucleotide sequence of the Bg_9562 gene has been submitted to NCBI under Accession no: KX620741.

**Protein expression and purification.** The complete CDS of Bg_9562 gene (333 bp) was PCR amplified from B. gladioli strain NGJ1 using Bg_9562fF1 and Bg_9562fR1 gene specific primers and cloned into pET28a bacterial expression vector (https://www.addgene.org/vector-database/2565/) to obtain pGD1 plasmid. Upon sequence validation, the pGD1 was transformed into E. coli (BL-21 strain, DE3-codon + ) for recombinant protein production. 250 ml of secondary bacterial culture was prepared by inoculating 1 % of primary culture ($10^9$ cells/ml). 1 mM IPTG was added at $OD_{600}$ 0.6 and incubated at 37 °C for 4 h with continuous shaking in 200 rpm. The bacterial pellet was sonicated in 30 ml buffer (10 mM PBS; pH 7.4, 1 mM lysozyme, and 1 mM Phenylmethanesulfonyl fluoride (PMSF)). The soluble fraction was incubated overnight at 4 °C with $Ni^{2+}$-NTA beads (pre-equilibrated with 10 mM PBS; pH 7.4). Thereafter beads were transferred in column and flow through was collected by washing the beads with 20 mM, 50 mM, and 70 mM imidazole. The recombinant Bg_9562 protein was eluted with linear gradient of 100–250 mM imidazole in a buffer containing 10 mM PBS (pH 7.4). Further western blot was performed by electro blotting protein onto polyvinylidene fluoride (PVDF) membrane and probed with mouse polyclonal anti-His-antibody (1:1000 dilutions). The presence of ~13 kDa band suggested overexpression and purification of the desired protein. The protein was identified by nanoLC-MS/MS using an Eksigent ekspert nanoLC 425 system coupled to AB Sciex TripleTOF 6600 System. For data analysis, the raw files were converted in to Mascot Generic Format (MGF) using msconvert and searched against the UniProt, NCBI, and common MS contaminant database using Mascot 2.5 (Matrix Science) software to identify the protein. The protein concentration was estimated by Bradford method[50].

**Expression dynamics of Bg_9562 through western blot analysis.** 2–3 days pre-grown R. solani mycelia were treated with NGJ1 cultures ($10^9$ cells/ml). After 48 h of co-cultivation, the mycelia were harvested and washed thoroughly with milli Q water to remove loosely attached bacteria. Total protein from the treated mycelia was isolated by crushing in liquid $N_2$. The powder was dissolved in 30 ml of buffer (10 mM PBS; pH 7.4, 1 mM lysozyme, and 1 mM PMSF) and upon centrifugation the supernatant was used as protein extract. Similarly as control, total protein was isolated from NGJ1 bacterial pellet as well as untreated R. solani mycelia. 10 μg of each protein was resolved on SDS PAGE gel (15%). Further the gel was electro blotted onto PVDF membrane and probed with polyclonal antibodies raised in rabbit against purified Bg_9562 protein. Western blot analysis was performed using anti-Bg_9562 (1:50,000 dilution) antibody as primary and alkaline phosphatase conjugated anti-rabbit IgG (Sigma) as secondary antibody (1:10,000 dilution) as per manufacturer's protocol (Sigma). Purified Bg_9562 protein was used as positive control in the blotting process. Further the ability of T3SS mutant bacterial strains (NGJ12 and NGJ13) to synthesize and translocate Bg_9562 protein into fungal mycelia was tested. For this, crude proteins were isolated from the fungal mycelia treated with both the T3SS mutants as well as from bacterial pellet of the mutants. Western blot analysis was performed as described above and experiment was repeated at least three times with freshly purified proteins of each sample.

**Testing efficacy of Bg_9562 protein on R. solani growth.** 300 μl of different concentrations of protein (5, 10, and 15 μg/ml) were used to treat R. solani sclerotia ($n = 5$) for 24 h at 28 °C. Equal number of sclerotia were also treated with various control solutions, i.e, heat inactivated Bg_9562 protein (by incubating 15 μg/ml of protein in boiling water for 45 min), 10 mM PBS (buffer in which the protein was dissolved) and 50 mM column wash. After treatment, each sclerotium was transferred onto fresh PDA plate to grow. The fungal growth for both protein as well as different control treated samples was measured in terms of area of mycelial lawn observed after 24 h, 48 h, and 72 h of incubation. Experiments were repeated at least three times with minimum three technical replicates and reproducible data were obtained.

Further efficacy of Bg_9562 protein on 48 h pre-grown R. solani mycelia was studied on agar slides. 300 μl of Bg_9562 protein (15 μg/ml) or PBS buffer (as control) was added individually onto the mycelia and slides were incubated aseptically at 28 °C to monitor the effect of protein treatment on mycelial growth. The agar slides were analyzed under Nikon 80i-epi-microscope to examine the effect on hyphal structure. Also, viability of protein-treated R. solani mycelia were studied through MTT staining. For this, 48 h pre-grown fungal mycelia were individually treated with 300 μl of protein (15 μg/ml) and buffer (10 mM PBS, pH 7.4) for 12 h at 28 °C. Further they were subjected to MTT assay as described above. The experiments were repeated at least three times with three technical repeats.

**Testing antimicrobial property of Bg_9562 protein.** Well-plate diffusion method was used to test the antifungal (S. cerevisiae and C. albicans) and anti-bacterial (B. gladioli, P. ananatis, and E. coli DH5α) activities of Bg_9562 protein. For this, the test microorganisms were plated onto their respective medium by spreading 50 μl of overnight culture. Thereafter wells were made aseptically at the center of each plate using sterile micro tip (1 ml) and 100 μl of the protein

(15 μg/ml) as well as buffer (10 mM PBS, pH 7.4) solutions were individually added into the wells. After overnight incubation at their respective growth conditions, inhibition zones of each microorganism were recorded.

For testing antifungal activity on several other fungi (Supplementary Table 3), the mycelial disks (obtained from fresh plates using 1 ml sterile micro tip) were treated with Bg_9562 protein (15 μg/ml) and buffer (10 mM PBS, pH 7.4) solutions for 24 h at 28 °C. Subsequently the treated as well as control mycelial disks were transferred onto fresh PDA plates and incubated at their respective growth temperatures. The fungal growth was measured in terms of area of mycelial lawn observed at 24 and 48 h of post treatment. The experiment was repeated at least three times with minimum three technical repeats.

**Deletion and complementation of Bg_9562 in NGJ1.** Partial fragment of Bg_9562 gene (209 bp) was PCR amplified from B. gladioli strain NGJ1 using Bg_9562pF2 and Bg_9562pR2 gene specific primers and cloned into pK18 mob vector[44] to obtain pGD2 plasmid and transformed into S17-1. Further the pGD2 was mobilized from S17-1 into NGJ2 (a spontaneous rif[r] derivative of B. gladioli strain NGJ1) through conjugation as per the method described earlier[45]. The insertion mutants were selected on kanamycin (50 μg/ml) containing KBA (King's medium B Base; Himedia, India) plates. Two independent single recombinant insertion mutants namely NGJ100 and NGJ101 were validated through PCR. The PCR with full-length gene specific primers (Bg_9562fF1 and Bg_9562fR1) failed to amplify the gene in both mutants while specific amplification was observed in case of wild type (NGJ2). Also, the PCR with Bg_9562pF2 (as forward primer) and M13F (as reverse primer) produced band in case of mutants but not in wild type. For complementation, full-length CDS (333 bp) of Bg_9562 gene was PCR amplified from NGJ1 by using Bg_9562cfF3 and Bg_9562cfR3 primers and was further cloned into pHM1 vector to obtain pGD3 plasmid. The pGD3 plasmid was electroporated (Gene pulsar Xcell[Tm]; BioRad) into NGJ101 strain to obtain NGJ102. Supplementary Table 4 summarizes list of bacterial strains, plasmids, and primer sequences used in this study.

**Mycophagous ability of Bg_9562 mutant and its complement.** Different concentrations ($10^9$ and $10^5$ cells/ml) of NGJ101 (Bg_9562 mutant) and NGJ102 (Bg_9562 complement) along with NGJ2 (rif[r] derivative of NGJ1) were used to treat R. solani sclerotia. After 4 h, the sclerotia were transferred onto PDA plates and incubated at 28 °C to monitor the confrontation.

**Heterologous expression of Bg_9562 in R. solanacearum.** R. solanacearum strain F1C1[34] and F1C1N2 (A hrpB⁻ and HR deficient mutant of F1C1) cultures were initially grown in BG media[51] and 100 μl of each culture was further inoculated in 10 ml minimal medium (g l⁻¹: FeSO₄–7H₂O, $1.25 \times 10^{-4}$; (NH₄)₂SO₄, 0.5; MgSO₄–7H₂O, 0.05; KH₂PO₄, 3.4; pH adjusted to 7 with KOH) containing 2 ml 50% glycerol. After 24 h of growth at 28 °C, 100 μl of the bacterial culture were mixed with 1 μg of plasmid (pGD3) and placed on nitrocellulose membrane kept on the top of BG-agar plate. After 48 h of co-cultivation, bacterial cells were harvested in sterile water and 100 μl of them were plated on spectinomycin (50 μg/ml) containing BG-agar plates. The transformed bacterial colonies were confirmed by PCR using Bg_9562 gene specific primers as well as vector specific primers. The pGD3 containing F1C1 and F1C1N2 were named as F1C1N3 and F1C1N4, respectively. The mycophagous interactions of different R. solanacearum strains with R. solani were tested as per the methodology used for B. gladioli strains. Further the growth rate of different R. solanacearum strains were studied in PDB broth containing pre-grown R. solani mycelia. For this, 1% of the bacterial culture ($10^9$ cells/ml) was inoculated in R. solani containing PDB broth and bacterial growth was estimated after 48 h of co-cultivation at 28 °C, by dilution plating on BG-agar plates. The experiment was repeated at least three times with minimum three technical repeats.

**Statistical analysis.** One-way analysis of variance was performed using Sigma Plot 12.0 (SPSS, Inc. Chicago, IL, USA) with $P \leq 0.001$ and $P \leq 0.05$ considered statistically significant. The statistical significance is mentioned in the figure legend, wherever required.

**Data availability.** The nucleotide sequence of the Bg_9562 gene of B. gladioli strain NGJ1 has been deposited in the NCBI nucleotide database under accession code KX620741. The authors declare that all other relevant data supporting the findings of the study are available in this article and its Supplementary Information files, or from the corresponding author upon request.

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

## Acknowledgements

I.T. and S.G. acknowledge fellowship from CSIR, Govt of India, while S.K.Y. and J.D. acknowledge fellowship from DBT, Govt. of India. We thank R.V. Sonti for providing E. coli strains, S.K. Ray for providing Ralstonia solanacearum F1C1 and its T3SS deficient mutant (F1C1N2), N. Tuteja for providing pET28a vector and T. R. Sharma for Magnaporthe oryzae strain. We sincerely thank R.V. Sonti, S.K. Ray, and Imran Siddiqi for comments on the manuscript. This work was supported by core research grant from National Institute of Plant Genome Research, India. Also, the research funding from DBT, Government of India to support the GJ lab is gratefully acknowledged.

## Author contributions

G.J. planned and supervised the experiments. G.J. and R.K. did initial identification and characterization of NGJ1; I.T. and R.K. carried out bacterial–fungal interaction analysis. D.M.S. carried out the protein related experiments and demonstrated the role of prophage tail protein in bacterial mycophagy. S.K.Y. carried out bacterial mutagenesis, complementation assays, phage related analysis, performed in depth microscopic analysis, and has been instrumental in revising the manuscript. S.K.Y., R.K., and J.D. had contributed towards characterizing the mycophagous interaction of different R. solanacearum strains. S.G. contributed in computational analysis and plant infection assays. D.M.S., S.K.Y., I.T., R.K., and G.J. contributed in manuscript writing and all authors approved the manuscript. The funders had no role in study design, data collection and analysis, decision to publish, or preparation of the manuscript.

## Additional information

**Competing interests:** A provisional patent application (no: 201611024726) entitled "Novel protein against fungal pathogens" has been filed in India to secure potential application of the Bg_9562 as a broad-spectrum antifungal molecule. D.M.S., S.K.Y., and G.J. are inventors in this patent application. The remaining authors declare no competing financial interests.

