## [Peer Review File · Nature Communications]

Reviewers' comments:

Reviewer #1 (Remarks to the Author):

This manuscript reveals novel findings about the mechanisms which enable certain bacteria to feed on fungi, namely the expression of a phage tail protein. A mutant of *Burkholderia gladioli* with an interrupted phage tail protein gene failed to feed on *Rhizoctonia solani* and complementation restored the wild-type phenotype. Moreover, exposure to purified phage protein alone was sufficient to mimic the strain's effect. Treatment of plants with the wild-type protected them against *R. solani*-induced disease, while the mutant did not. While the findings of this study are novel and in my opinion potentially of broad interest to the scientific community, I see several shortcomings which should be addressed by the authors:

- Hardly any quantitative data are presented. The vast majority of results is presented in form of Petri dish or microscopic pictures of which the reader must believe that they are representative, although the readability of these pictures is very poor. The pictures should illustrate quantitative data showing means of the different experiments conducted, their variability and their statistical significance; e.g. LL47-49, what the authors describe cannot be seen on the pictures
- The authors argue that the findings could result in producers using phage proteins for plant protection against fungal diseases. To this end, such a phage protein should have a specific effect on phytopathogens and not a broad effect on various fungi as seems to be the case here (ext. data fig. 5 and 10). Those fungi that could not be tested for susceptibility to mycophagy by *B. gladioli* because of strong mycelial inhibition should have been tested for susceptibility to the phage protein, as this would indicate whether the effect is of specific or of broad nature. Oomycetes, which have different cell walls and are important disease-causing agents too, should be tested as well, as they might react differently to the phage protein.
- The material and methods is very poorly described and does not allow replication of the experiments (one example: LL12-15 extended data: what was the age of *R. solani* when used for confrontation, what was the age and the OD of the bacterial culture, etc.)
- Minor comments : all organisms used should be characterized (extended data L 8); extended data table 1: I would calculate the difference between before and after growth/exposure and only test those values for statistical significance; ext. data fig 1 and many others: pictures are distorted and one cannot see what one should see on such poor pictures (see also above, need for quantitative data); bacterial abundance should be quantified with harvest/dilution/plating and CFU counting, not merely by appearance; ext. data fig. 5: it seems that *B. gladioli* was not inoculated in the same way for all fungi; L114: did the authors try to integrate the broad host range plasmid carrying the phage protein in non mycophagous bacteria to see if they become mycophagous? To me this would be an important additional proof of the importance of the phage protein in mycophagy; fig. 2: fluorescently tagged bacterial strains should be used for better visualization of the interaction between bacteria and fungi.

Reviewer #2 (Remarks to the Author):

The manuscript by Swain et al is an account of the deleterious effect of a *Burkholderia gladioli* strain NGJ1 encoded protein [related to bacteriophage tail as well as hypothetical proteins] on the rice-pathogenic fungus *Rhizoctonia solani*. The authors show that the protein encoded by the gene located in the context of a prophage on the *B. gladioli* genome is indeed mycolytic and may be involved in the antifungal nature of *B. gladioli*.

The manuscript contains a large suite of experimental data that together bring a thought-provoking and potentially interesting story.

However, the descriptions in the text are not rigorous, and there are numerous examples of unclear or sloppy formulations. Moreover, the data are often rather qualitative in nature (e.g. l. 34: several hundreds.; l. 35: ..only a few..; many other instances), where they should be more quantitative.

The supernatant of *R. solani* apparently could feed *B. gladioli*, but the authors are not clear about 'conditions'. In other words, do they mean to say that the fungus needs to be 'triggered' to release compounds, or does the fungus by itself releases such compounds? And, what compounds are involved? Such information is needed to complete the story.

The identification of a phage tail like protein by a type 3 secretion system (T3SS) effector prediction program is surprising and raises the hypothesis that the T3SS mediates its release. However, this attractive hypothesis is neither tested nor discussed further. Does *B. gladioli* contain such a system, and of which subtype? Didn't the authors – with their approach - set out to identify such effectors that are injected into fungal mycelium by the T3SS?

The identified protein cannot be rightly called a phage tail protein unless its function as such is proven [see also comment below].

A main critique is that the paper invokes more questions than it answers. The mycolytic activity of the protein [predicted by the T3SS effector prediction program], encoded by a gene that is located in the middle of an apparent prophage, raises the immediate hypothesis that a whole bacteriophage may be utilized by *B. gladioli* in its antagonism against *R. solani*. Thus the way gene expression is triggered, and whether prophage induction is involved, are key issues to be addressed. However, the authors do not provide any data or hypotheses in this respect. A second issue is the question whether the identified prophage is common across *B. gladioli* strains, or whether it is unique for the used strain NGJ1. Moreover, do the (many) hypothetical proteins identified in the tree also occur in a prophage genetic background? If so, this would strengthen the tenet that such a context is necessary for the activity.

Specific comments:

Title: the title is wrong. It should read: ...feed on fungi [instead of ...feed fungi]

l. 37-38: this description is too qualitative. A more concrete description is needed.

l. 41 on: this needs more details about the treatment, time, temperature, conditions, etc.

l.48: drastically increased: this requires data, i.e. quantitation.

l.49: ..bacterial cells are associated...: how many are associated, how many are not, what time course of development is followed?

l.59/60: how is this experiment controlled? And how are the data quantified and replications used?

l.79/80: the tree should be better discussed. Apparently, the levels of homology to phage tail proteins or to hypothetical proteins were very low. What other hits were found? Were there any hints at other types of function? Did the other genes also occur in the context of a prophage sequence? Such are issues that need to be explored before one can firmly state that a phage tail protein is encoded. Moreover, to make this statement, proof should be given for protein function as a phage tail constituent!

l.92-97: these data are too qualitative. Quantitation is needed.

l.103-105: what fungi were not inhibited, and what could be the reason of that?

l. 113-114, and before: the involvement of the identified protein is likely. However, the authors did not detail any difficulties they may have had with respect to heterologous expression, folding, formation of inclusion bodies, etc. More information on these technical aspects is required.

Extended tab1 1: the variation is very high. What are the significances of the differences?

Extended data Fig 1: unclear; indicate what is what on the plate. Also indicate how sclerotia are visualized

Extended data Fig 2: unclear; indicate in (a) how mycelium is differentiated from sclerotia, and in (b) how this should be interpreted (three replicates of what?).

Extended data Fig. 3: it is understood that the left panel was infested with just *R. solani*. Correct? A time course of development is needed here. The lower panel shows affected tomato leaves on the left [but apparently the legend is wrong]. How can this be explained if *B. gladioli* was present? And, is the control rightly described [no microorganisms?]

Extended data Fig. 4: what inoculum was used? How does this grow in e.g. LB? *E. coli* did not grow, but how was this controlled? In other words, are the compounds present in the supernatant of *R. solani* truly specific for *B. gladioli*?

Extended data Fig 5: although often used, showing such plates is not rigorous. At least, a time

course of development versus proper controls is required.

Extended data Fig. 9: what do the arrows indicate? It is unclear what alterations the authors mean here. More concrete description needed.

Responses to referees comments:

Reviewer#1:

Comment: This manuscript reveals novel findings about the mechanisms which enable certain bacteria to feed on fungi, namely the expression of a phage tail protein. A mutant of *Burkholderia gladioli* with an interrupted phage tail protein gene failed to feed on *Rhizoctonia solani* and complementation restored the wild-type phenotype. Moreover, exposure to purified phage protein alone was sufficient to mimic the strain's effect. Treatment of plants with the wild-type protected them against *R. solani*-induced disease, while the mutant did not. While the findings of this study are novel and in my opinion potentially of broad interest to the scientific community, I see several shortcomings which should be addressed by the authors.

Response: We are very much thankful for your comments. We have revised the MS to remove various shortcomings as per your suggestions.

Comment: Hardly any quantitative data are presented. The vast majority of results is presented in form of Petri dish or microscopic pictures of which the reader must believe that they are representative, although the readability of these pictures is very poor. The pictures should illustrate quantitative data showing means of the different experiments conducted, their variability and their statistical significance; e.g. LL47-49, what the authors describe cannot be seen on the pictures

Response: Thank you for pointing this out. We realized that in-deed our MS was lacking quantitative data. In this revised MS, we have tried our best to provide quantitative data wherever required and also provided data to reflect variability and their statistical significance (Figure 4, Extended Data Figure 1, 4, 5, 11, 15 and 18). Further we have also replaced several of the figures to improve their readability.

Line 47-49, we have now included the data obtained by counting the colony forming units (CFU), suggesting enhanced growth of bacteria during bacterial mycophagy (Extended Data Figure 1).

Comment: The authors argue that the findings could result in producers using phage proteins for plant protection against fungal diseases. To this end, such a phage protein should have a specific effect on phytopathogens and not a broad effect on various fungi as seems to be the case here (ext. data fig. 5 and 10). Those fungi that could not be tested for susceptibility to mycophagy by *B. gladioli* because of strong mycelial inhibition

should have been tested for susceptibility to the phage protein, as this would indicate whether the effect is of specific or of broad nature. Oomycetes, which have different cell walls and are important disease-causing agents too, should be tested as well, as they might react differently to the phage protein.

Response: Earlier we have presented data that the NGJ1 has very strong antifungal activity on some of the fungi and due to strong mycelial inhibition mycophagy couldn't be demonstrated on them. However as suggested by you, when we adopted uniform bacterial inoculation pattern [by spotting 20µl of bacterial culture (10^9 cells/ml) at four different corners of petri dish while fungi were grown in centre], we observed NGJ1 to demonstrate mycophagy on all tested fungi (Extended Data Figure 6).

As kindly suggested by you, we also tested the effect of Bg_9562 protein on different fungi, including oomycetes fungal pathogen *Phytophthora* sp. and observed broad spectrum antifungal activity (Extended Data Figure 14 and 15).

Comment: The material and methods is very poorly described and does not allow replication of the experiments (one example: LL12-15 extended data: what was the age of *R. solani* when used for confrontation, what was the age and the OD of the bacterial culture, etc.).

Response: As suggested we have now elaborated the material and methods section to allow replication of various experiments. Age of *R. solani* and OD of bacterial inoculum, etc. have been provided in the revised methods.

Minor comments

Comment: All organisms used should be characterized (extended data L 8);

Response: As per your suggestion, we have now classified the uncharacterized fungi up to genus level (Extended Data Table 3).

Comment: extended data table 1: I would calculate the difference between before and after growth/exposure and only test those values for statistical significance

Response: Thank you for the kind suggestions. As suggested we have calculated the difference between initial biomass and mass obtained after 48h of growth exposure and further tested their statistical significance (Extended Data Figure 5).

Comment: Ext. data fig 1 and many others: pictures are distorted and one cannot see what one should see on such poor pictures (see also above, need for quantitative data); bacterial abundance should be quantified with harvest/dilution/plating and CFU counting, not merely by appearance;

Response: Thank you for the kind suggestion. We realised that Extended Data Figure 1 was becoming distorted. Hence we have distributed the data in two different figures (Extended Data Figure 2 and 3). In order to depict variation amongst different repeats that NGJ1 is capable of preventing sclerotial growth and demonstrates mycophagous activity even at lower dilutions, we have now included Extended Data Table 2.

Further as suggested, the bacterial abundance during mycophagous development of NGJ1 on rich and minimal media was estimated through colony counting. Data is summarised in Extended Data Figure 1.

Comment: Ext. data fig. 5: it seems that *B. gladioli* was not inoculated in the same way for all fungi;

Response: Thank you for pointing this out. As per your kind suggestion, we have now adopted a uniform inoculation pattern [by spotting 20µl of bacterial culture (10^9 cells/ml) at four different corners of petri dish while the fungi were grown in centre] and interestingly we could observe mycophagy on all tested fungi, including *Phytophthora* sp. (Extended Data Figure 6).

Comment: L114: did the authors try to integrate the broad host range plasmid carrying the phage protein in non mycophagous bacteria to see if they become mycophagous? To me this would be an important additional proof of the importance of the phage protein in mycophagy;

Response: Thank you for the kind suggestions. As suggested we integrated the pHM1 containing Bg_9562 in a non mycophagous *Ralstonia solanacearum* strain F1C1. The results revealed that the recombinant *R. solanacearum* becomes mycophagous in a functional T3SS dependent manner (Extended Data Fig 17, Extended Data Fig. 18). This brings additional proof of importance of phage tail like protein in mycophagy and also supports the potential T3SS secreted effector nature of this protein.

Comment: fig. 2: fluorescently tagged bacterial strains should be used for better visualization of the interaction between bacteria and fungi.

Response: We did tag the bacteria with GFP and used it to study mycophagous interaction with *R. solani*. But due to very strong auto-fluorescence of *R. solani* in GFP filter under confocal microscope, we were unable to unambiguously detect the bacterial cells. Considering this limitation, in this study we focused on light microscopic analysis.

Reviewer #2:

Comment: The manuscript by Swain et al is an account of the deleterious effect of a *Burkholderia gladioli* strain NGJ1 encoded protein [related to bacteriophage tail as well as hypothetical proteins] on the rice-pathogenic fungus *Rhizoctonia solani*. The authors

show that the protein encoded by the gene located in the context of a prophage on the *B. gladioli* genome is indeed mycolytic and may be involved in the antifungal nature of *B. gladioli*. The manuscript contains a large suite of experimental data that together bring a thought-provoking and potentially interesting story.

Response: Thank you very much for your kind comments.

Comment: However, the descriptions in the text are not rigorous, and there are numerous examples of unclear or sloppy formulations.

Response: As suggested we have now elaborated the text throughout the length of the revised MS. We sincerely hope that we have presented our points clearly this time.

Comment: Moreover, the data are often rather qualitative in nature (e.g. l. 34: several hundreds.; l. 35: ..only a few..; many other instances), where they should be more quantitative.

Response: Thank you for your kind suggestions. In this revised MS, we have now included several quantitative data to reflect variation across different experiments/repeats (Figure 4, Extended Data Figure 1, 4, 5, 11, 15 and 18). For example with respect to line 34, 35; we have now included a table (Extended Data Table 1) to show that only few sclerotia are produced on an NGJ1 confrontation plate.

Comment: The supernatant of *R. solani* apparently could feed *B. gladioli*, but the authors are not clear about 'conditions'. In other words, do they mean to say that the fungus needs to be 'triggered' to release compounds, or does the fungus by itself releases such compounds? And, what compounds are involved? Such information is needed to complete the story.

Response: Thanks for the kind suggestions. We realize that this would be another interesting area of research to understand whether the NGJ1 triggers release of compounds from fungi or fungus by itself releases such compounds. However, in this revised MS, we rather focused on understanding mechanistic insight of bacterial mycophagy and present the novel and unanticipated role of phage tail like protein during mycophagy.

Please note that we had used the cell free mycelial extract to show that NGJ1 but not *E. coli* is capable of utilizing the *R. solani* extract to grow. However we realized the data presented in this regard, was not adding much mechanistic insight about mycophagy and hence we have preferred not to include it in this revised MS.

Comment: The identification of a phage tail like protein by a type 3 secretion system (T3SS) effector prediction program is surprising and raises the hypothesis that the T3SS mediates its release. However, this attractive hypothesis is neither tested nor

discussed further. Does *B. gladioli* contain such a system, and of which subtype? Didn't the authors – with their approach - set out to identify such effectors that are injected into fungal mycelium by the T3SS?.

Response: Thank you for the kind comment. It is in-deed very surprising that a phage tail like protein of NGJ1 is having potential type 3 secretion system (T3SS) signal. As suggested, we have modified the MS to show that NGJ1 contains an injectosome subtype of T3SS (Extended Data Figure 7). We isolated two independent T3SS mutants of NGJ1 and observed both of them to exhibit defects in mycophagy, suggesting the important of T3SS in bacterial mycophagy (Extended Data Fig. 8).

The western blot analysis revealed that the protein is synthesized by the bacterium and during interaction it is translocated into *R. solani*, as we have detected the Bg_9562 in NGJ1 treated *R. solani* mycelial protein extract (Extended Data Fig. 12). Furthermore, we are now presenting data to demonstrate that the expression of Bg_9562 gene imparts mycophagous activity in otherwise non-mycophagous bacterium *Ralstonia solanacearum* strain F1C1 in a functional T3SS dependent manner (Extended Data Fig. 17 and Extended Data Fig. 18). Taken together these results suggested that the Bg_9562 is potentially a T3SS effector and T3SS dependent delivery in fungi is required for its mycophagous activity.

Comment: The identified protein cannot be rightly called a phage tail protein unless its function as such is proven [see also comment below].

Response: We agree with the reviewer. As the protein has high sequence similarity with other phage tail proteins (Extended Data Figure 9), we felt it would be more appropriate to refer this protein as phage tail like protein. We have accordingly revised the MS.

Comment: A main critique is that the paper invokes more questions than it answers.

Response: We do agree that the paper raises more questions than it answers. However, we feel that the findings are novel, unexpected and represent a major advanced in our knowledge of this field. We are indeed excited about this and the results can be explored in future studies.

Comment: The mycolytic activity of the protein [predicted by the T3SS effector prediction program], encoded by a gene that is located in the middle of an apparent prophage, raises the immediate hypothesis that a whole bacteriophage may be utilized by *B. gladioli* in its antagonism against *R. solani*. Thus the way gene expression is triggered, and whether prophage induction is involved, are key issues to be addressed. However, the authors do not provide any data or hypotheses in this respect.

Response: Thank you for the kind suggestion. In spite of various efforts (phage plaque assay on soft agar plate, potential phage particle isolation followed by scanning electron microscopic analysis, PCR using cell free supernatant with a few bacterial house-keeping genes as well as genes present in *Bg_9562* cluster) we do not observed any prophage induction.

As the phages are generally evolved to utilize bacterial transcriptional and translational machinery, we feel that as such there would be no advantage for the bacteriophage to feed on fungi. Hence to avoid any further focus on phages, we preferred not to include these data as they might distract readers from the main focus of the present study. To avoid further confusion, we are now referring the *Bg_9562* as phage tail like protein and concentrated on characterizing its role with respect to mycophagy.

In this revised MS, we are now presenting data that the protein is synthesized by NGJ1 and is delivered into *R. solani* mycelia during confrontation (Extended Data Fig. 12). Further the expression of the gene imparts mycophagous ability in otherwise non-mycophagous *Ralstonia solanacearum* in a functional T3SS dependent manner (Extended Data Fig 17, 18).

Comment: A second issue is the question whether the identified prophage is common across *B. gladioli* strains, or whether it is unique for the used strain NGJ1. Moreover, do the (many) hypothetical proteins identified in the tree also occur in a prophage genetic background? If so, this would strengthen the tenet that such a context is necessary for the activity.

Response: Thank you for this valuable suggestion. As suggested, we have analysed the available genomes of different *B. gladioli* strains and observed that identified prophage locus is common amongst them (Extended Data Figure 10). Also we observed that several of the hypothetical proteins identified in the tree also occur in a prophage genetic background, although a few of them are associated with in-complete phage cluster (Extended Data Figure 9).

Specific comments

Comment: Title: the title is wrong. It should read: ...feed on fungi [instead of ...feed fungi]

Response: Thanks for the kind suggestion. As suggested we have modified the title as "A phage tail like protein is deployed by a bacterium to feed on fungi".

Comment: I.37-38: this description is too qualitative. A more concrete description is needed.

Response: Thank you for pointing this out. As suggested, we have now provided quantitative data to show the impact of bacterial treatment on sclerotial germination and variation across different replicates (Extended Data Table 2). Further we have improved quality of Extended Data Figure 2 and Extended Data Figure 3.

Comment: I.41 on: this needs more details about the treatment, time, temperature, conditions, etc.

Response: As suggested we have now provided details of treatment, time, temperature and conditions in the methods section (Extended Data Page 2, L35-41).

Comment: I.48: drastically increased: this requires data, i.e. quantitation.

Response: As suggested we are now providing bacterial quantification data in Extended data Figure 1.

Comment: I.49: bacterial cells are associated...: how many are associated, how many are not, what time course of development is followed?

Response: Thanks for the kind comment. A fluorescently tagged bacterium would have helped us to estimate how many bacteria are associated with fungi. However, due to strong auto-fluorescence of *R. solani* in GFP filter, we are not able to unambiguously count the GFP signal of tagged bacteria. Hence we restricted ourselves to light microscopic analysis to show that significant number of bacteria is associated with fungi.

Further as suggested, we followed the time course of mycophagous interaction of NGJ1 with *R. solani* and have modified Figure 2 to depict the interaction at different time points (24h, 48h and 72h).

Comment: I.59/60: how is this experiment controlled? And how are the data quantified and replications used?

Response: Thank you for pointing this. Please refer to our response to your related question, we realized that the data was not adding much value to focus of the revised MS; hence we have removed this data.

Comment: I.79/80: the tree should be better discussed. Apparently, the levels of homology to phage tail proteins or to hypothetical proteins were very low. What other hits were found? Were there any hints at other types of function? Did the other genes also occur in the context of a prophage sequence? Such are issues that need to be explored before one can firmly state that a phage tail protein is encoded. Moreover, to make this statement, proof should be given for protein function as a phage tail constituent!

Response: Thank you for the kind suggestions. BlastX analysis (Extended Data Figure 9) showed the gene to have homology with different phage tail proteins of bacteria as well as some bacteriophages (*B. gladioli* phage KS14, *Ralstonia* phage RSA1 and *Yersinia* phage L-413c). No other hits were found.

The phylogenetic and blastX analysis further reflected very high levels of homology to phage tail proteins of different *Burkholderia gladioli* strains (Extended Data Figure 9). Further significant homology was observed amongst Bg_9562 orthologs of different *Burkholderia* and *Paraburkholderia* species. Also we have now highlighted that several Bg_9562 orthologs are apparently present in context of prophage. Hence considering all this bioinformatics analysis, we have named this Bg_9562 as phage tail like protein and accordingly modified our MS.

Comment: I.92-97: these data are too qualitative. Quantitation is needed.

Response: We have now included quantitative data related to MTT assay and effect of protein treatment on fungal growth in Extended Data Figure 11b and Figure 4b, respectively.

Comment: I.103-105: what fungi were not inhibited, and what could be the reason of that?

Response: Thank you for the comment. We have tested various fungi (Extended Data Table 3) and observed that the protein has broad spectrum antifungal activity on all of them (Extended Data Figure 14; Extended Data Figure 15).

Comment: I.113-114, and before: the involvement of the identified protein is likely. However, the authors did not detail any difficulties they may have had with respect to heterologous expression, folding, formation of inclusion bodies, etc. More information on these technical aspects is required.

Response: Thank you for the comment. The protein was overexpressed in *E. coli* and purified from soluble fractions, without much difficulty. We have included details of protein expression and purification in the method section of this revised MS.

Comment: Extended tab1 1: the variation is very high. What are the significances of the differences?

Response: We have calculated the difference between initial biomass and mass obtained after 48h of growth exposure and tested their statistical significance (Extended Data Figure 5).

Comment: Extended data Fig 1: unclear; indicate what is what on the plate. Also indicate how sclerotia are visualized.

Response: Thank you for pointing this out. In order to avoid confusion, we have now indicated how sclerotia and mycelia are visualized on petri plate (Extended Data Figure 2). The sclerotia are spore like resting structures of *R. solani* and upon germination they give rise to mycelia which at the end of growth period again differentiate to form secondary sclerotia.

Please note that to improve the quality of images we have modified Extended Data Figure 1 as two separate figures (Extended Data Figure 2, Extended Data Figure 3). Further we have included Extended Data Table 2 to summarise the quantitative effect of bacterial treatment on sclerotial growth.

Comment: Extended data Fig 2: unclear; indicate in (a) how mycelium is differentiated from sclerotia, and in (b) how this should be interpreted (three replicates of what?).

Response: As we clearly mentioned in Extended Data figure 2 about how sclerotia and mycelia are differentiated, we preferred not to indicate them again in this figure (now Extended Data Figure 5a). The figure reflects that the bacterial treatment prevented germination of *R. solani* sclerotia. Photographs provided in this figure are representative of images obtained from independent repeats. Further weight reduction of *R. solani* caused due to NGJ1 treatment and the reproducibility across three independent replicates is being summarised as Extended Data Figure 5b.

Comment: Extended data Fig. 3: it is understood that the left panel was infested with just *R. solani*. Correct? A time course of development is needed here. The lower panel shows affected tomato leaves on the left [but apparently the legend is wrong]. How can this be explained if *B. gladioli* was present? And, is the control rightly described [no microorganisms?]

Response: Thank you for pointing out issues with the legend (now Extended Data Figure 4). We realized that the legend was very confusing. Hence we have modified the figure and legend to make it clear. As suggested, we are now presenting the images and disease severity index caused by *R. solani* (with and without NGJ1 treatment) on rice, at two different time points (3 and 5 dpi).

For tomato infection, we have presented data at 3 dpi only, as by 5 dpi, the entire leaves become necrotic. We hope that it would be clear now that the treatment of NGJ1 prevents disease caused by *R. solani* in both rice as well as tomato. While *R. solani* sclerotia (control; without NGJ1 treatment) caused characteristics disease symptoms.

Comment: Extended data Fig. 4: what inoculum was used? How does this grow in e.g. LB? *E. coli* did not grow, but how was this controlled? In other words, are the compounds present in the supernatant of *R. solani* truly specific for *B. gladioli*?

Response: Although it seems that the compounds present in *R. solani* extract are specifically promoting the growth of *B. gladioli*, we have realized that the data might be confusing and it is not adding much insight about bacterial mycophagy; hence we have deleted this data in this revised MS.

Comment: Extended data Fig 5: although often used, showing such plates is not rigorous. At least, a time course of development versus proper controls is required.

Response: Thank you for kind suggestion. We have realized that the Extended Data Figure 5 was not very clear. As suggested we are now presenting data at two different time points (3 and 7 dpi) so that one can clearly visualize mycophagous development of NGJ1 on different fungi (Extended Data Figure 6). Please note that at 3 dpi the mycophagy is not initiated while it is clearly visible at 7 dpi.

Comment: Extended data Fig. 9: what do the arrows indicate? It is unclear what alterations the authors mean here. More concrete description needed.

Response: Thank you for pointing this out. We have now indicated alterations caused due to bacterial or protein treatment in *R. solani* in the figure legend and indicated them with arrows (Extended Data Figure 13)

Reviewers' comments:

Reviewer #1 (Remarks to the Author):

I found the MS substantially improved and I am satisfied with the authors' response to my initial comments. Going through the MS and the extended data, I found some minor points to be corrected, which I am listing below:

Comments on main text:

L62: spell out MTT when first used

L71: to grow in dune soil

L77: as well as the oomycete *Phytophthora* sp., which has a non-chitinaceous cell wall.

Comments on extended data:

L11-12 : how were sclerotia collected ? description is missing here.

L16 : plating (check throughout)

L19, although (...) bacteria were found

L21 : bacteria treated

L28 : were PDA plates supplemented with an antifungal compound to allow counting bacterial CFU without fungal growth? Or was the rif-resistant derivative used? Please specify.

L32: what is "semi-dry" weight? Fresh weight? (concerns also y axis label of extended fig. 5)

L65: spell out MTT also when first using the term in extended data.

L166: concentrations

L176: or PBS buffer (check throughout)

L183: at least

L187 vs. L191: micro-organisms or microorganisms, but consistent

L231: plated

L306: remove "visual" (or replace by visible, or physical)

I would combine extended data figs. 2 and 3

Check throughout: into vs. in to

Extended fig. 18: data should be plotted in log scale.

Reviewer #2 (Remarks to the Author):

By the title, this is an interesting piece of work. The authors have obtained evidence for the tenet that a protein-encoding gene that is sitting in a prophage region in *Burkholderia gladioli* has antifungal activity. Moreover, it appears to be excreted by a type III secretion system.

However, there are consistent flaws in the development of the story that would appear to prevent the publishing of the current text. This reviewer is uncertain how much of the perceived lack of depth of the data or interpretations, ambiguous or imprecise descriptions, lack of quantitation is due to a general poor use of the English language, or to true problems in the study.

The authors need to consult an expert proficient in scientific writing in English, to take out the multiple problems in the text; these are simply too numerous to list here.

Moreover, the text is overlong and repetitive.

There is a lack of information as to the completeness of the identified phage (can it be induced to form a phage population?). And, the hits with other proteins/genes are unclear: are these all similar-length proteins? Also, did the authors do a domain analysis, so as to assess whether a particular domain of the 111-aa protein would be akin to a known toxic or lytic domain.

The rationale for picking the 111-aa proteins from amongst 35 proteins is missing. Or, was this just a 'lucky shot'? Would any of the other proteins also have fungal-killing activity?

The detection of the protein from *R. solani* mycelia is uncertain, as the fungus was in contact with the bacterium, which thrived on it (l. 100-102). How was this controlled?

Overall, it seems weird that the bacterium blocks sclerotium germination but consumes fungal hyphae. The authors do not address this issue, which is of prime ecological importance.

A range of major other concerns follow (written out as questions or doubts):

- l. 15: mechanistic insights underlying .. are unknown. This is not correct, as there are major advances in bacteria thriving on fungi. Check the definition of mycophagy.
 - l. 40: treatment of what? (possibly, a number of sclerotia - how many? - placed on agar)
 - l. 49: CDA? write out what this contains. Is it a minimal medium?
 - l. 56: significant number: this is a term from statistics, please give the number, or a range. Compared to what?
 - l. 62: MTT: explain first time of appearance in text
 - l. 68-69: repetitive with before
 - l. 84,85: suddenly a T3SS mutant comes up; this needs introduction earlier on. How it was made, tested, etc, deserves description, as it is crucial for the interpretations.
 - l. 104: it appears that the 15 ug/ml does not represent the level per unit fungal biomass; the latter is crucial.
 - l. 115: glass slide, versus (earlier) agar slide. How sterile was this, what was in the agar (nutrients)?
 - l. 130: plasmid integration: what plasmid (type, size), how achieved (selection), was it checked for double inserts, etc. This is a too loose statement.
 - l. 161 on: as there is no evidence of phage propagation, this is speculative.
- Extended data 1 through 18: there are numerous problems/concerns:
- Fig 1: CFU quantification: difficult and not quantitative. b" legend: cells instead of CFUs?
 - Fig 2: inhibition is not clearly separable from competition in this assay.
 - Fig. 3: Difficult to see; cells/ml, how much on the plate?
 - Fig. 7: is this T3SS canonical? Any gene missing? Compare to another proven T3SS.
 - Fig. 9: indicate the Bg protein (middle? now general description)
 - Fig 13: control R. sol. (untreated) is missing; must be added.
 - Fig 18: were the initial R. solani levels similar across all treatments? This is necessary to make a fair comparison. Does PDB support growth of the bacteria?

Reviewers' comments:

Reviewer #1 (Remarks to the Author):

Comment: I found the MS substantially improved and I am satisfied with the authors' response to my initial comments. Going through the MS and the extended data, I found some minor points to be corrected, which I am listing below:

Response: Thank you very much for appreciating our revised MS. As suggested we have modified our MS to correct all minor points.

Comments on main text:

Comment: L62: spell out MTT when first used

Response: As suggested we have spelled out the MTT as 3-(4,5-dimethylthiazol-2-yl)-2,5-diphenyltetrazolium bromide (Line no 68-69).

Comment: L71: to grow in dune soil

Response: We have modified the sentence to emphasize that the *Collimonas* sp has ability to grow in presence of fungal mycelia (Line no 76-78). The sentence now reads as “For example, the *Collimonas* sp. exhibits enhanced growth in presence of *Absidia* sp. as well as common dune soil fungi such as *Chaetomium globosum*, *Fusarium culmorum*, *Mucor hiemalis*”

Comment: L77: as well as the oomycete *Phytophthora* sp., this has a non-chitinaceous cell wall.

Response: We have modified the sentence as per your suggestion (Line no 85).

Comments on extended data:

Comment: L11-12 : how were sclerotia collected ? description is missing here.

Response: We have now mentioned in the method section (Extended Data) that we have used sterile toothpicks to collect sclerotia from the fungal plates (Extended Data_Line no 13-15).

Comment: L16 : plating (check throughout)

Response: We have corrected it throughout the MS

Comment: L19, although (...) bacteria were found

Response: We have now modified the sentence and shifted it to main text of the MS to emphasize that observed sclerotial growth inhibition is due to bacterial treatment (main MS_Line no 41-45).

Comment: L21 : bacteria treated

Response: corrected.

Comment: L28 : were PDA plates supplemented with an antifungal compound to allow counting bacterial CFU without fungal growth? Or was the rif-resistant derivative used? Please specify.

Response: As *B. gladioli* strain NGJ1 is itself antifungal, we did not supplement PDA plates with any antifungal compound for counting bacterial CFU. Please note that similar results were obtained when rif-resistant derivative (NGJ2) was used.

Comment: L32: what is "semi-dry" weight? Fresh weight? (concerns also y axis label of extended fig. 5)

Response: Please note that the pre-grown fungal mycelia as well as bacterial treated fungal mycelia were blotted on sterile filter disc to remove excess water to measure the fungal weight. As we had to use it for subsequent growth assay (to understand the effect of bacterial treatment) we did not completely dry the mycelia. Considering this, we preferred to use the term semi-dry weight in our MS. We have now elaborated these details in the method section (Extended Data_Line no 31-36).

Comment: L65: spell out MTT also when first using the term in extended data.

Response: as suggested we have provided the full form of MTT (Extended Data_Line no 72).

Comment: L166: concentrations

Response: corrected

Comment: L176: or PBS buffer (check throughout)

Response: corrected

Comment: L183: at least

Response: corrected

Comment: L187 vs. L191: micro-organisms or microorganisms, but consistent

Response: We have used microorganisms at both the places

Comment: L231: plated

Response: corrected

Comment: L306: remove "visual" (or replace by visible, or physical)

Response: we have replaced "visual" with 'physical'

Comment: I would combine extended data figs. 2 and 3

Response: As suggested we have now combined both the figures and referring it as Extended Data Fig. 2.

Comment: Check throughout: into vs. in to

Response: We have checked 'into' and 'in to' and corrected their usages

Comment: Extended fig. 18: data should be plotted in log scale.

Response: As suggested, we have plotted data in log scale (now Extended Data_Fig. 17).

Reviewer #2 (Remarks to the Author):

Comment: By the title, this is an interesting piece of work. The authors have obtained evidence for the tenet that a protein-encoding gene that is sitting in a prophage region in *Burkholderia gladioli* has antifungal activity. Moreover, it appears to be excreted by a type III secretion system.

Response: Thank you for your kind appreciation.

Comment: However, there are consistent flaws in the development of the story that would appear to prevent the publishing of the current text. This reviewer is uncertain how much of the perceived lack of depth of the data or interpretations, ambiguous or imprecise descriptions, lack of quantitation is due to a general poor use of the English language, or to true problems in the study.

Response: We are thankful for your kind comments on the write up which helped us to express more correctly. Earlier we tried to be precise in main text of the MS and elaborated technical/methodological details in the extended data. We realized that this was creating imprecise description in the main text of our MS, leading to difficulty in interpretation of data. As per your valuable suggestion, we have now thoroughly revised the MS to remove ambiguities and made it easy to be followed by the broader readers.

Comment: The authors need to consult an expert proficient in scientific writing in English, to take out the multiple problems in the text; these are simply too numerous to list here. Moreover, the text is overlong and repetitive.

Response: Thank you for pointing this. We have taken help from our senior colleagues and tried our best to improve the English language.

Comment: There is a lack of information as to the completeness of the identified phage (can it be induced to form a phage population?).

Response: After carefully analysing the bacteriophage locus in different *Burkholderia gladioli* strains, we observed that the locus is incomplete. We have now

clearly mentioned that the locus lacks phage head assembly/capsid proteins in the main text of our revised manuscript (Line no 110-112 of the main text; Extended Data Fig 9). Overall this suggests that locus cannot be induced to form active prophage. This is supported by our observations that in spite of various efforts (phage plaque assay on soft agar plate, potential phage particle isolation followed by scanning electron microscopic analysis, PCR with cell free supernatant using a few bacterial house-keeping genes as well as genes present in *Bg_9562* cluster), phage induction was not observed.

It is noteworthy that we demonstrated that *Bg_9562* protein is synthesized by NGJ1 and it gets delivered into fungal mycelia in a T3SS dependent manner. Further we also observed that heterologous expression of *Bg_9562* protein in *Ralstonia solanacearum* imparts mycophagous ability and a functional T3SS is required for gain of such activity. Taken together, these results confirm that the phage induction is not required for mycophagous ability.

Comment: And, the hits with other proteins/genes are unclear: are these all similar-length proteins? Also, did the authors do a domain analysis, so as to assess whether a particular domain of the 111-aa protein would be akin to a known toxic or lytic domain.

Response: Earlier, Extended Data Fig. 9 was depicting phylogenetic relationship and BLAST search scores, reflecting the sequence similarity of *Bg_9562* protein with phage tail proteins of other bacteria/phages. As suggested, we have analysed the amino acid sequence of various such proteins and found them to be approximately of similar size (pl see the modified Extended Data Fig. 8). Further we have studied the domain structure of *Bg_9562* protein and found that it does not contain any known toxic or lytic domain. The *Bg_9562* protein as well as its various orthologs were found to harbour phage_TAC_7 domain, generally present in bacteriophage phage tail protein Gp41. We have now updated these information in the main text (Line no 104-107) as well as Extended Data Fig. 8 of this modified MS.

Comment: The rationale for picking the 111-aa proteins from amongst 35 proteins is missing. Or, was this just a 'lucky shot'? Would any of the other proteins also have fungal-killing activity?

Response: As mentioned in our revised MS, the presence of potential T3SS signal in *Bg_9562* protein showing homology to phage tail proteins was quite surprising. Hence in this study, we focused on characterizing this protein with respect to bacterial mycophagy. We would focus on characterizing the role of other potential T3SS secretion proteins in our future study.

Comment: The detection of the protein from *R. solani* mycelia is uncertain, as the fungus was in contact with the bacterium, which thrived on it (l. 100-102). How was this controlled?

Response: As suggested in order to control the experiment, we have now included data with both wild type (NGJ1) as well as T3SS mutants (NGJ12 and NGJ13). Data clearly reflects that the protein is detected from the NGJ1 treated *R. solani* mycelia but is not detected from the mycelia treated with either of the T3SS mutants (Main text Line no 118-123). Overall this suggests that during confrontation the bacteria translocate the protein into *R. solani* mycelia in a functional T3SS dependent manner (Please see Extended Data Fig. 11).

Comment: Overall, it seems weird that the bacterium blocks sclerotium germination but consumes fungal hyphae. The authors do not address this issue, which is of prime ecological importance.

Response: Our study demonstrates that at higher concentration, the bacterium blocks sclerotium germination while at lower concentration it consumes fungal hyphae. This suggests that at higher concentration the antifungal activity of the bacterium is prominent while at lower concentration the mycophagous activity is apparent. We speculate that in natural habitat/ecosystem, concentration/population of the bacterium would not be high enough to suppress sclerotial germination while it can still facilitate mycophagous activity at low concentration. This in turn would be advantageous for the bacterium, as it can utilize fungal biomass to support/sustain its growth.

A range of major other concerns follow (written out as questions or doubts):

Comment: -l. 15: mechanistic insights underlying .. are unknown. This is not correct, as there are major advances in bacteria thriving on fungi. Check the definition of mycophagy.

Response: Thank you for pointing this out. We do agree that involvement of cell wall degrading enzymes and toxins such as tolaasin, syringomycin etc had been previously shown to be associated with mycophagy. We have cited a classical review article on this topic which has covered various known aspects of bacterial mycophagy (Leveau and Preston, 2007). Considering this we have modified our sentence to lower down the tone (Line no 15, 16).

Comment: -l. 40: treatment of what? (possibly, a number of sclerotia - how many? - placed on agar)

Response: Thank you for pointing this out. Please refer to Extended data table 2 and method section (Line no 39-46), we have used 5 sclerotia for bacterial treatment and have incubated them on PDA plates for studying their subsequent growth. Please note that this experiment has been repeated several times.

Comment: -l 49: CDA? write out what this contains. Is it a minimal medium?

Response: CDA is a sort of minimal medium used for fungal cultivation. As suggested we have now mentioned the composition of CDA in method section (Extended data; Line no 8-10).

Comment: -l. 56: significant number: this is a term from statistics, please give the number, or a range. Compared to what?:

Response: Please note that similar question has been raised by the previous reviewer also. As mentioned in our earlier response, fluorescently tagged bacterium would have helped us to estimate how many bacteria are associated with fungi. However, due to strong auto-fluorescence of *R. solani* under GFP filter, we are not able to unambiguously count the GFP signal of tagged bacteria. Hence we have to restrict ourselves to light microscopic analysis. Considering that significant number reflects statistical term being associated with comparison, we have replaced 'significant number' with 'large number' in this modified MS (Line no 63-64). We intend to depict that large number of NGJ1 bacteria are associated with fungal mycelia during confrontation.

Comment: -l. 62: MTT: explain first time of appearance in text

Response: As suggested we have elaborated MTT as 3-(4,5-dimethylthiazol-2-yl)-2,5-diphenyltetrazolium bromide, first time of appearance in text (Line no 68-69)

Comment: -l. 68-69: repetitive with before

Response: We have modified the sentence in abstract as well as main text of the MS. As we are not separately writing the introduction, we feel that it is required to re-introduce the mycophagy term in the main body of our MS, to conclude that the *B. gladioli* strain NGJ1 has mycophagous property.

Comment: -l. 84,85: suddenly a T3SS mutant comes up; this needs introduction earlier on. How it was made, tested, etc, deserves description, as it is crucial for the interpretations.

Response: Thank you for pointing this out. As mentioned, we are not separately writing the introduction section in our MS. The first section of our MS is focused on demonstrating the mycophagous property of *B. gladioli* strain NGJ1. In the second section, we have focused on characterizing the mechanistic detail and involvement of T3SS in bacterial mycophagy. Hence we are introducing the T3SS at this place and have modified the sentences for better connectivity.

Further considering your valuable suggestion, in this modified MS, we have briefly mentioned how the T3SS mutants were raised and tested for mycophagy (Line no 91-94 in the main text) The technical details about how the T3SS mutant was made, tested etc. had been described in the method section of Extended Data; Line no 111-119).

Comment: -l. 104: it appears that the 15 ug/ml does not represent the level per unit fungal biomass; the latter is crucial.

Response: Thanks for pointing this out. We have used protein for sclerotial treatment and observed that 15 µg/ml of purified protein was sufficient in preventing sclerotial germination (text in the MS has been modified accordingly; Line no 127-128). To make it further clear, we have now modified method section by providing the details of protein treatment (Extended Data; Line no 181-182).

Comment: -l. 115: glass slide, versus (earlier) agar slide. How sterile was this, what was in the agar (nutrients)?

Response: Thanks for pointing this out. We mean glass slides containing 1% agar. The glass slides were made sterile by autoclaving and the autoclaved 1% agar was spread over it to make the agar slides (Please see the modified method section in Extended Data; Line no 47-49). Please note that beside agar, the slides do not contain any other nutrients.

Comment: -l. 130: plasmid integration: what plasmid (type, size), how achieved (selection), was it checked for double inserts, etc. This is a too loose statement.

Response: Thanks for pointing this out. The details of the plasmid integration to generate gene mutant and testing for its proper integration had been described in the method section (Extended Data; Line no. 216-233). Please note that similar phenotype was observed with two independent mutants and phenotype was fully complemented by expressing the full length copy of the gene on broad host range plasmid, pHM1. Hence there was no necessity to check the double inserts. As per your suggestion, we have now mentioned the name of the plasmid used for integration in the main text as well (Line no 154).

Comment: -l. 161 on: as there is no evidence of phage propagation, this is speculative.

Response: We have deleted this sentence now.

Comment: Extended data 1 through 18: there are numerous problems/concerns:

Fig 1: CFU quantification: difficult and not quantitative. b" legend: cells instead of CFUs?

Response: Most humbly I wish to state that counting colony forming unit (CFU) of the bacteria is most precise and widely used methodology to quantify the bacterial abundance. Generally for bacterial colonies, the term CFU is preferred and hence we wish to keep the term CFU in the figure legend.

Comment: Fig 2: inhibition is not clearly separable from competition in this assay.

Response: Thanks for pointing this out. As suggested by referee 1, we have now combined the Extended Data Fig 2 and 3. It is apparent that at higher concentration the bacteria is preventing germination of fungal sclerotia while upon treatment with lower concentration of bacteria, the sclerotia could initially germinate but subsequently bacteria grew over it and prevented further growth of the fungi.

One can argue that inhibition is not clearly separable from the competition in this assay. To rule out such possibility, we have performed similar assay on acidified PDA plates also. Please refer to the main figure 1b; the control sclerotia (without bacterial treatment) could germinate on acidified PDA plates, while the bacterial treated sclerotia failed to germinate in spite the fact that there was no bacteria which surround the sclerotia. Taken together, our data demonstrates that the NGJ1 is capable of preventing the germination of *R. solani* sclerotia at higher concentration while at lower dilution it demonstrates mycophagy.

We have now elaborated this in the modified text of the MS (Line no 45-51).

Comment: Fig. 3: Difficult to see; cells/ml, how much on the plate?

Response: Please note we have now combined the figure 2 and 3. To make it clear, we wish to state that in this assay we have treated fungal sclerotia with different concentrations of bacterial cells (mentioned in brackets) for 4h and grew them on PDA plates at 28⁰C (Please refer to method section; Extended Data; Line no 40-41).

Comment: Fig. 7: is this T3SS canonical? Any gene missing? Compare to another proven T3SS.

Response: Yes this is a canonical, non-flagellar type T3SS. We have used an online tool (<http://bacterial-virulence-factors.cbgp.upm.es/T346Hunter>) to analyse whether the locus harbour complete T3SS and observed that all the core component of canonical T3SS is conserved at this locus. The data is presented in modified Extended Data Figure 6.

Comment: Fig. 9: indicate the Bg protein (middle? now general description)

Response: As suggested we have indicated the Bg protein along with its general description in the phylogenetic tree. Further we have mentioned that a conserved phage_TAC_7 domain is present in various orthologs (Please see modified Extended Data Fig.8)

Comment: Fig 13: control *R. sol.* (untreated) is missing; must be added.

Response: As suggested, we have modified the figure and have included control *R. solani* microscopic picture (Please see modified Extended Data Fig.12).

Comment: Fig 18: were the initial *R. solani* levels similar across all treatments? This is necessary to make a fair comparison. Does PDB support growth of the bacteria?

Response: Thanks for pointing this out. In this assay we have analysed the growth of Bg_9562 containing wild type *Ralstonia solanacearum* (F1C1N3) and T3SS deficient mutant (F1C1N4) in PDB broth with or without presence of *R. solani* mycelia. For fair comparison, we had used similar initial concentrations of *R. solanacearum* across various treatments. We observed that PDB does not support the growth of *R. solanacearum* while presence of fungal mycelia facilitated the growth of Bg_9562 expressing *R. solanacearum* in a T3SS dependent manner. To make it more apparent, we have now included data at two different time points (12h and 48h). Further as suggested by referee 1, we have now plotted the data in log scale (Please see modified Extended Data Fig.17).